# Illustrating new understanding of adsorbed water on silica for inducing tetrahedral cobalt(II) for propane dehydrogenation

Zijun Huang[1,2,3,4], Dedong He [1,2,3,4] ✉, Weihua Deng[1,2,3], Guowu Jin[2,3,4], Ke Li[2,3,4] & Yongming Luo [1,2,3,4] ✉

Highly dispersed metal sites on the surface of silica, achieved from immobilization of metal precursor within hydroxyl groups, has gained increasing attention in the field of heterogeneous catalyst. However, the special role of adsorbed water derived by hydroxyl groups on the silica is generally ignored. Herein, a new understanding of adsorbed water on the formation of highly dispersed tetrahedral Co(II) ($T_d$-cobalt(II)) sites is illustrated. It is indicated that sufficient adsorbed water induces the transformation of precursor of $Co(NO_3)_2$ into intermediate of $[Co(H_2O)_6]^{2+}$. Subsequently, $[Co(H_2O)_6]^{2+}$ makes the highly dispersed $T_d$-cobalt(II) sites to be available during direct $H_2$-reduction process. A systematic characterization and DFT calculation prove the existence of the adsorbed water and the importance of the intermediate of $[Co(H_2O)_6]^{2+}$, respectively. The as-synthesized catalyst is attempted to the propane dehydrogenation, which shows better reactivity when compared with other reported Co based catalysts.

Supported metal catalysts are known as one of the most important candidates for heterogeneous catalysis[1]. In the vast majority of catalytic processes, the interaction between support and active metal site is of great importance in determining their catalytic performance[2]. When utilizing reducible metal oxides as supports[3–5], taking advantage of their reducibility and capacity for metal-support bonding, the concept of strong metal-support interaction is proposed[6–8]. In terms of non-reducible supports of $Al_2O_3$ and $SiO_2$, these interactions can hardly involve significant charge transfer or the involvement of support lattice oxygen in the catalytic cycle[9,10]. Hydroxyl groups on the surface of these supports are therewith emphasized as the bridge to build required interaction between metal and support, which is beneficial to acquiring highly dispersed and stable metal sites on the supports[11,12].

The Al-OH sites on $Al_2O_3$ are revealed for anchoring metal species. Abundant hydroxyl groups on the γ-$Al_2O_3$ cause the single-atom dispersion of Ag species[13]. Nevertheless, coordinatively unsaturated aluminum atoms on the $Al_2O_3$ also shows an anchoring effect for metal sites[14]. From this point of view, silica-based support serves as an ideal candidate to independently study the interaction between hydroxyl groups and active metals. Vodyankina and Bao reported that the presence of surface OH groups on the $SiO_2$ regulated the distribution and size of Ag nanoparticles[12,15]. More recently, Smith et al. found that local distribution of OH groups on SBA-15 stabilized V species and contributed to preferable catalytic behaviors[16]. Taken together, in silica-supported metal catalysts, surface hydroxyl plays a vital part in regulating the dispersion of the introduced metal sites. In general, the adsorbed water is commonly preserved in conventional synthesis process, and the participation of both water molecules and the derived hydroxyl groups in heterogeneous catalysis is fundamental and important. However, the special role of adsorbed water on the silica is generally ignored, resulting in the lack of a deeper understanding of synthesis mechanism.

[1]Faculty of Environmental Science and Engineering, Kunming University of Science and Technology, Kunming 650500, P. R. China. [2]Faculty of Chemical Engineering, Kunming University of Science and Technology, Kunming 650500, P. R. China. [3]The Innovation Team for Volatile Organic Compounds Pollutants Control and Resource Utilization of Yunnan Province, Kunming 650500, P. R. China. [4]The Higher Educational Key Laboratory for Odorous Volatile Organic Compounds Pollutants Control of Yunnan Province, Kunming 650500, P. R. China. ✉e-mail: dedong.he@qq.com; environcatalysis@kust.edu.cn

In the meantime, with the growing demand for propylene and the explosion of the shale gas revolution[17], propane dehydrogenation (PDH) process has become more commercially attractive[18,19], which exhibits significant environmental and energy implications. However, commercial PDH processes using high cost of $PtSn/Al_2O_3$ and high toxicity of $Cr/Al_2O_3$ catalysts have restricted their further developments[20,21]. Recently, the well-defined and highly dispersed sites of Fe[22], Ga[23], Zn[24], Co[25] and Y[26], obtained via grafting tailored metal precursors within hydroxyl on the silica supports, have shown the capability to break C-H bonds of alkanes. Among these catalysts, highly dispersed tetrahedral Co(II) sites ($T_d$-cobalt(II)) have gained much interest, due to its superior PDH catalytic potential[27,28]. Hock et al. developed strong electrostatic absorption (SEA) method to synthesize $T_d$-cobalt(II) contained $Co/SiO_2$ catalyst[27]. Similarly, Sooknoi et al. investigated the influence of different cobalt precursors, including $[Co(NH_3)_5Cl]Cl_2$, $[Co(bipy)_3](NO_3)_2$ and $[Co(en)_2Cl_2]Cl$, on the reactivity of the $Co/SiO_2$ catalyst[29]. Besides, surface organometallic chemistry (SOMC) was employed to obtain this highly dispersed $T_d$-cobalt(II) through protonolysis of organometallic precursor within surface OH groups on the $SiO_2$, and bulky ligands were devoted to facilitate sites isolation[25,29]. In these cases, although hydroxyl is fully utilized to realize the required interaction between metal and support, expensive metal precursors and complicated synthesis routes are necessary to achieve precise control over the surface structure of the catalysts[30]. Specifically, adsorbed water coexists normally with the surface hydroxyl groups of silica support, focuses towards the role of adsorbed water in regulating the generation of highly dispersed and stable metal sites, however, is yet underestimated.

Herein, we described the role of adsorbed water during the synthesis of highly dispersed $T_d$-cobalt(II) sites onto the silica support. The results showed that the hydrates were critical to the tetrahedral Co site formation. It was proposed that under a simple direct reduction process, the adsorbed water could help to immobilize the Co precursor, and then further form $T_d$-Co(II) sites. A systematic characterization and DFT calculation proved the existence of the adsorbed water and the importance of the intermediate of $[Co(H_2O)_6]^{2+}$, respectively. More importantly, the resulting $T_d$-cobalt(II)/SBA-15 catalyst was superefficient for PDH reaction, which exhibited better reactivity when compared with other reported Co based catalysts. The present work illustrates new understanding of adsorbed water on silica for inducing the formation of highly dispersed cobalt(II) sites, and provides simple and effective approach to design high reactivity of cobalt-based PDH catalyst.

## Results

### Formation of tetrahedral Co(II)

We primarily attempted to compare the difference in the chemical states of cobalt species obtained from direct reduction (Dir-reduction/ catalyst precursor was directly reduced by $H_2$) and indirect reduction ($H_2$-reduction/air-calcination sample was reduced by $H_2$) processes. Firstly, XRD was examined to identify the crystal structure of Co species on different samples (Fig. 1a). The diffraction peaks of Air-calcination sample fit well with that of the standard $Co_3O_4$ at $2\theta = 36.9°$, $59.3°$ and $65.2°$ (JCPDS No. 42-1467). After $H_2$ reduction (indirect reduction), the peaks of $Co_3O_4$ disappeared and a weak diffraction peak of metallic Co at $2\theta = 41.2°$ was observed, which was further confirmed by TEM from Supplementary Fig. 1, where obvious aggregation of Co with interplanar spacing of $d_{(103)} = 0.453$ nm were detected. In contrast, the Co species from the Dir-reduction catalyst were undiscerned (Fig. 1a). Additionally, from Fig. 1g and Supplementary Fig. 2-4 (TEM and STEM/EDS-mapping), the highly homogeneous dispersion of Co species was witnessed, and the SAED pattern indicated the amorphous Co states, suggesting the markedly improved dispersion of Co species via the $H_2$-direct reduction method. Secondly, XPS was used to investigate the surface chemical states of

catalysts. All the samples showed two broad and asymmetric main peaks at about 775–790 eV and 790–810 eV (Fig. 1b), which were corresponded to Co $2p_{3/2}$ and Co $2p_{1/2}$, respectively. For the Co $2p$ spectra of the Air-calcination catalyst, two spin-orbit doublet peaks of Co(II) and Co(III) and their broad satellite peaks were displayed. The dominating peaks of Co(II) at BE = 781.5 eV and Co(III) at BE = 779.4 eV were in good agreement with those reported for $Co_3O_4$[31]. Moreover, metallic Co evidenced by the BE = 778.1 eV was found in the $H_2$-reduction sample[32], suggesting that $Co_3O_4$ was reduced by $H_2$ treatment, and consistent with the XRD and TEM results. It is worth noting that all of the Co species in the Dir-reduction catalyst were existed as Co(II) (Co $2p_{3/2}$ at 781.8 eV and Co $2p_{1/2}$ at 797.8 eV), revealing that Co was neither reduced to $Co^0$ nor oxidized to $Co_3O_4$ on the Dir-reduction catalyst. Moreover, quasi in-situ XPS measurement from Supplementary Fig. 5 confirmed again that only Co(II) species were found on the Dir-reduction sample.

Following, $H_2$-TPR was performed to characterize the reduction behavior of the cobalt species (Fig. 1c). Air-calcination catalyst presented three reduction peaks at about 300 °C, 350 °C, and 500 °C, ascribing to the subsequent reduction from $Co_3O_4$ to metal Co ($Co^{3+} \rightarrow Co^{2+} \rightarrow Co^0$)[33]. No reduction peak was observed in the $H_2$-reduction catalyst, since the catalyst was in-situ reduced in the TPR equipment. Namely, $H_2$ treatment consumed the oxygen in $Co_3O_4$ to form metal Co, which can be supported by the XRD, XPS, and TEM results. Interestingly, very different reduction curve was exhibited on the Dir-reduction sample, and the main feature of the peaks in the TPR profiles was apparently shifted to high temperature region at approximately 800 °C, revealing that Co species were connected strongly within the silica support and hard to be reduced[34]. The UV-Vis spectra provided the configuration of various cobalt species. Adsorption bands at about 410 nm and 720 nm were observed on the Air-calcination catalyst, which were allocated to the ligand-metal charge-transfer of $O^{2-} \rightarrow Co^{2+}$ and $O^{2-} \rightarrow Co^{3+}$ in spinal $Co_3O_4$[35]. By contrast, $H_2$-reduction catalyst showed very weak peak due to $Co_3O_4$ was reduced into $Co^0$ phase. For the Dir-reduction catalyst, a broad absorption band with maximum peaks at approximately 543 nm, 578 nm, and 642 nm were observed, which was associated with $v_3$ ($^4A_2 \rightarrow {^4}T_1(P)$) transition characteristic of the tetrahedral Co(II) ($T_d$-Co(II))[36,37].

To acquire more information about the coordination and structural features of the Co species in the Dir-reduction catalyst, EXAFS measurements were implemented. The XANES spectrum of the Dir-reduction catalyst (red line) from Fig. 1e showed an absorption edge located between Co foil and CoO (closer to CoO). Therefore, combining the EXAFS and XPS data, we inferred that the valence state of cobalt in the Dir-reduction catalyst was around +2. Consistently, the conclusion of Bader charge calculation from Supplementary Fig. 6 also demonstrated that the oxidation state of Co was very close to +2. Besides, the results of EXAFS fitting and corresponding Fourier transform were summarized in Fig. 1f, Supplementary Fig. 7 and Supplementary Table 1. The EXAFS spectra recorded Co-O and Co-Co shells with a distance of 2.07 Å and 3.12 Å, respectively. The corresponding coordination number (CN) were 4.0 and 4.8, which was lower than the average number of the CoO phase, indicating that the tetracoordinated Co(II) species dominated over the Dir-reduction catalyst. It was clarified that the presence of cationic cobalt bonded to the silica with $Co-O-SiO_n$ linkages at the $Co-SiO_2$ interfaces stabilize the dispersed Co species[38].

Correspondingly, the discussed characterizations illustrated collectively that the $Co_3O_4$ was mainly existed in the Air-calcination catalyst, and it was easily reduced into metallic Co after $H_2$ reduction. Importantly, as for the Dir-reduction catalyst, highly dispersed and extremely stable $T_d$-Co(II) reduced hardly by $H_2$ flow below 800 °C were obtained. Besides, the formation of $T_d$-Co(II) species was also evidenced on the Dir-reduction catalysts with different Co contents (4 and 6%), as revealed from Supplementary Fig. 8 (XRD),

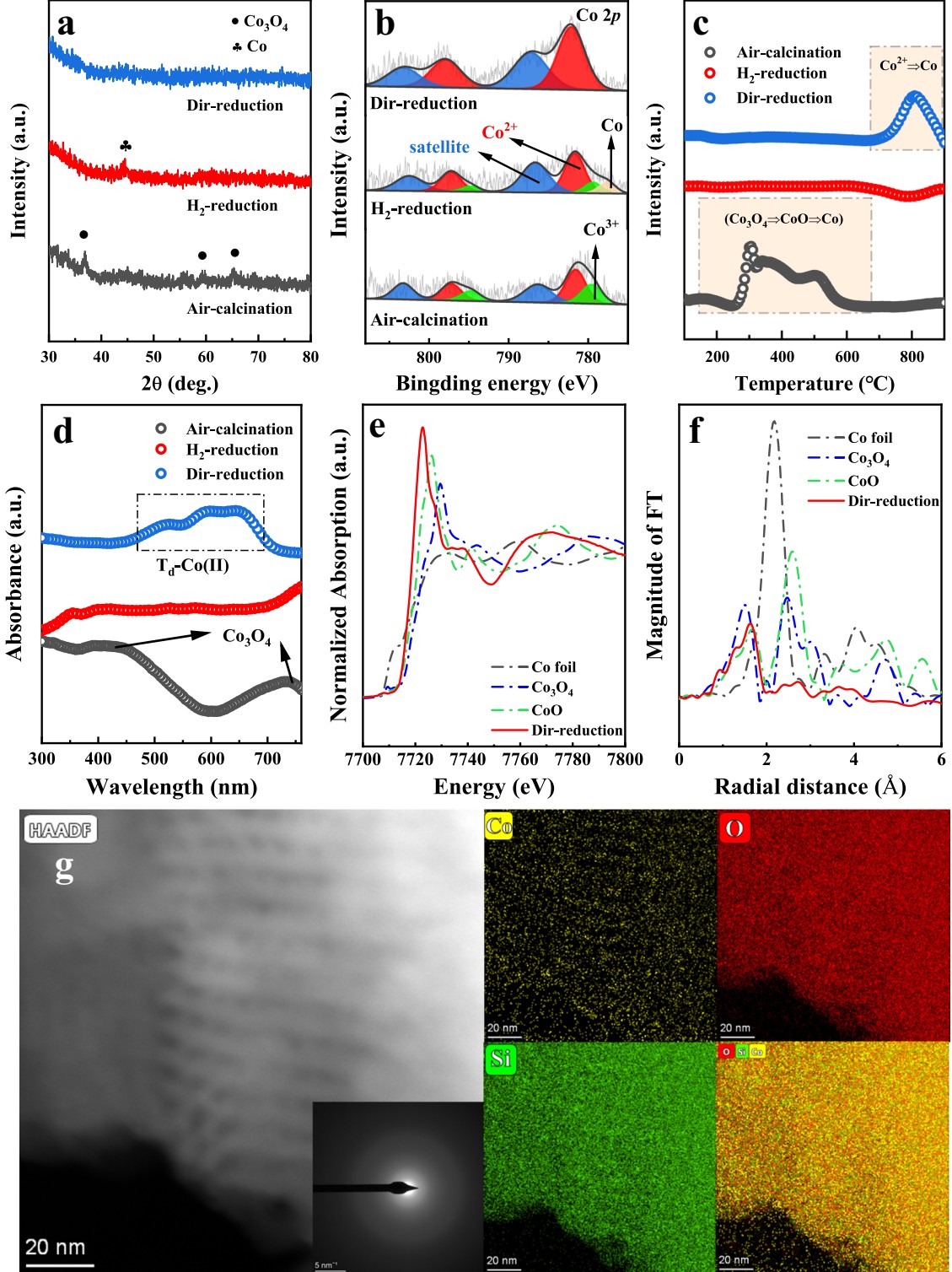

**Fig. 1 | Functional characterization of Co/SBA-15 catalysts. a** XRD patterns, **b** Co 2p XPS spectra, **c** $H_2$-TPR profiles, **d** UV-vis spectra for Dir-reduction, Air-calcination and $H_2$-reduction catalysts, **e** the normalized intensity of Co K-edge XANES spectra, **f** the corresponding Fourier transformation of $k^3$-weighted EXAFS oscillation, **g** STEM image and EDS elements mapping of Dir-reduction catalyst.

Supplementary Fig. 9 (XPS), Supplementary Fig. 10 ($H_2$-TPR), and Supplementary Fig. 11 (UV-vis).

## Verifying the effect of absorbed water on $T_d$-Co(II) formation

It is universally accepted that surface OH groups on $SiO_2$ are the anchoring sites to precisely acquire the isolated metals[39]. Nevertheless, the role of adsorbed water on silica is always neglected, since the stabilization of metal on supports occurs at high temperature. On account of this, following experiments were designed to demonstrate the importance of absorbed water in the formation of isolated $T_d$-Co(II) (Fig. 2 and Supplementary Fig. 12, it is noted that the four contrast samples of PM, PM200, PM900 and IMP900 were all conducted by $H_2$-direct reduction treatment, and the reduction temperature was 600 °C). Generally, in the process of catalyst synthesis, the source of

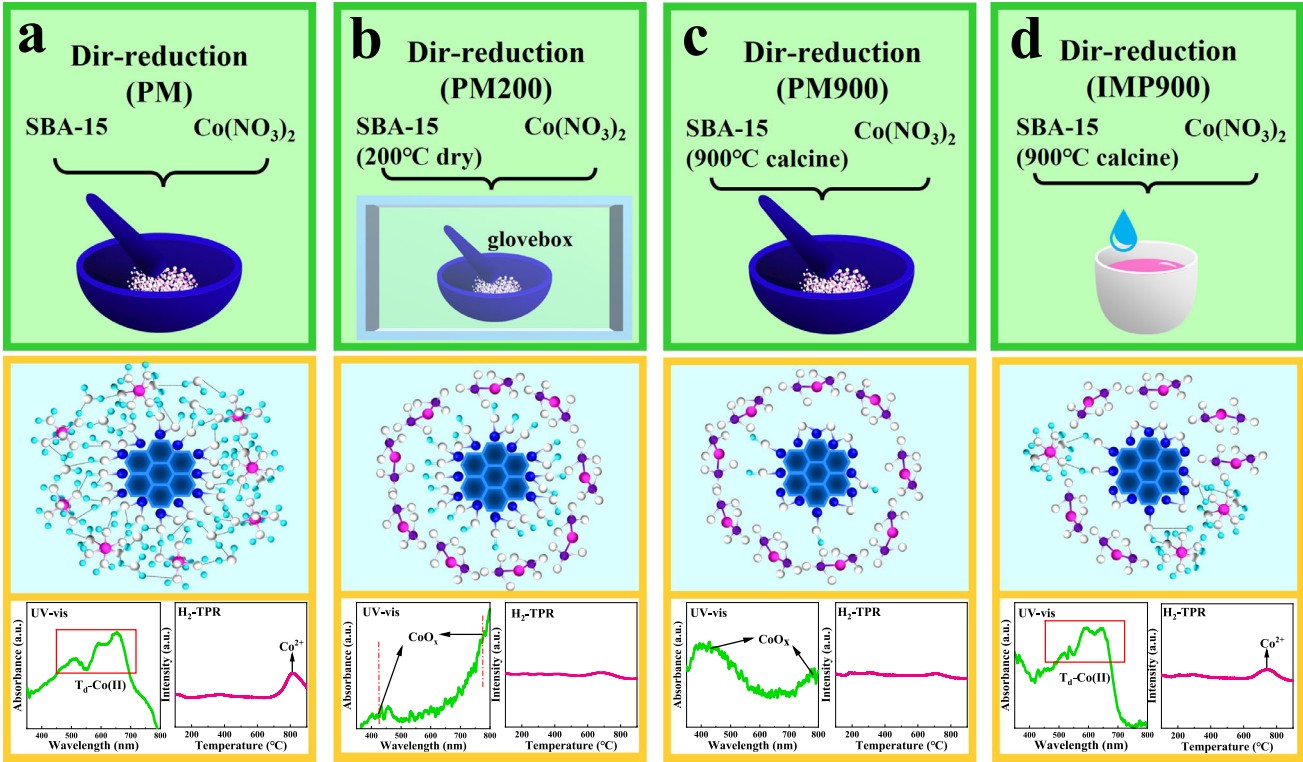

**Fig. 2 | Schematics of four design experiments were devoted to verify the effect of absorbed water. a** physically mix SBA-15 and cobalt precursor. **b** physically mix SBA-15(200 °C) and cobalt precursor. **c** physically mix SBA-15(900 °C) and cobalt precursor. **d** impregnate SBA-15(900 °C) within cobalt precursor.

the water was classified into two categories, one was the liquid water added during the impregnation process, and the other was the adsorbed water connected with OH groups on the support. Firstly, SBA-15 support was mixed physically with $Co(NO_3)_2 \cdot 6H_2O$ and subjected to $H_2$-direct reduction treatment (Dir-reduction (PM)), to exclude the effect of liquid water on the formation of Co species (Fig. 2a, none of liquid water was exposed). Secondly, the hydroxylated surface of SBA-15 with a predominance of silanol groups is hydrophilic in nature, which presented a large amount of adsorbed water on the support. To eliminate the influence of adsorbed water (Fig. 2b, neither liquid water nor adsorbed water was introduced), SBA-15 was heated at 200 °C (TG data in Supplementary Fig. 13 illustrated the complete removal of absorbed water), and then mixed physically with $Co(NO_3)_2 \cdot 6H_2O$ in glove box before conducting $H_2$-direct reduction (the sample was reduced directly by $H_2$, Dir-reduction (PM200)). Thirdly, most of the surface hydroxyl groups were removed irreversibly by calcining SBA-15 at 900 °C (Supplementary Fig. 14), although it still retained the characteristic structure of SBA-15 (Supplementary Fig. 15). Afterwards, 900 °C-calcined SBA-15 was mixed physically with $Co(NO_3)_2 \cdot 6H_2O$ and reduced directly by $H_2$ to obtain Dir-reduction (PM900) sample (Fig. 2c, without absorbed water and surface hydroxyl). Fourthly, the residual little amount of surface OH on the 900 °C-calcined SBA-15 was utilized to obtain small amount of adsorbed water during impregnating $Co(NO_3)_2 \cdot 6H_2O$ onto the support (the sample was reduced directly by $H_2$, Dir-reduction (IMP900), Fig. 2d, a small amount of absorbed water was presented). The details for the existence of adsorbed water and hydroxyl groups in the four design experiments were given in Supplementary Fig. 12.

Specifically, in Fig. 3a (XRD), no diffraction peak was seen in all the catalysts, implying uniform distribution of Co species. In Fig. 3b (in-situ TPR), Dir-reduction (PM) catalyst exhibited one significant peak higher than 800 °C, representing the reduction of Co(II) species that were strongly interacted with silica support. In contrast, almost no remarkable reduction peak was observed in the TPR curves of the Dir-

reduction (PM200) and the Dir-reduction (PM900) catalysts, suggesting the absence of Co(II) species that were intensely interacted with SBA-15 support. Significantly, in the Dir-reduction (IMP900) catalyst, the characteristic reduction peak of embedded Co(II) sites on the support was observed, but the temperature and the intensity of the reduction peak at around 750 °C was downshifted when compared to the Dir-reduction (PM) sample, revealing that Co species did not interact strongly with the support when the surface OH was insufficient. Furthermore, quasi in-situ XPS of the Dir-reduction (PM) catalyst from Fig. 3c showed the BE of typical high-spin Co(II) species. On the contrary, the rest three catalysts indicated the peaks of metallic Co. In addition, samples were analyzed by the ex-situ UV-vis and recorded in Fig. 3d. The absorbance at 543 nm, 578 nm, and 642 nm, as the characteristic band for $T_d$-Co(II), were observed in the Dir-reduction (PM) and Dir-reduction (IMP900) catalysts. But the intensity of $T_d$-Co(II) over the Dir-reduction (IMP900) was lower than that on the Dir-reduction (PM). Over the Dir-reduction (PM200) and Dir-reduction (PM900) catalysts, the characteristic peak representing for cobalt oxide at approximately 410 nm and 720 nm were detected. It is worth noting that in Fig. 3b there was no reduction peak below 600 °C can be found because these four catalysts were subjected to $H_2$-direct reduction treatment at 600 °C. However, since metallic cobalt was easily re-oxidized in air, ex-situ UV-vis spectra captured the diffraction peak of cobalt oxide (Fig. 3d). Therefore, it can be drawn that $T_d$-Co(II) can only be obtained in the Dir-reduction (PM) and Dir-reduction (IMP900) catalysts that contained adsorbed water. Namely, adsorbed water on the SBA-15 support was essential for the acquisition of required $T_d$-Co(II).

A detailed analysis of how Co species work with absorbed water will be discussed in the following. UV-vis spectra of the samples dried at 100 °C (without reduction treatment) were depicted in Fig. 3e. As for the catalyst precursors of Dir-reduction and Dir-reduction (PM) that contained sufficient adsorbed water, the existence of $[Co(H_2O)_6]^{2+}$ was supported by the typical triplet at 525 nm, 575 nm, and 650 nm[40,41].

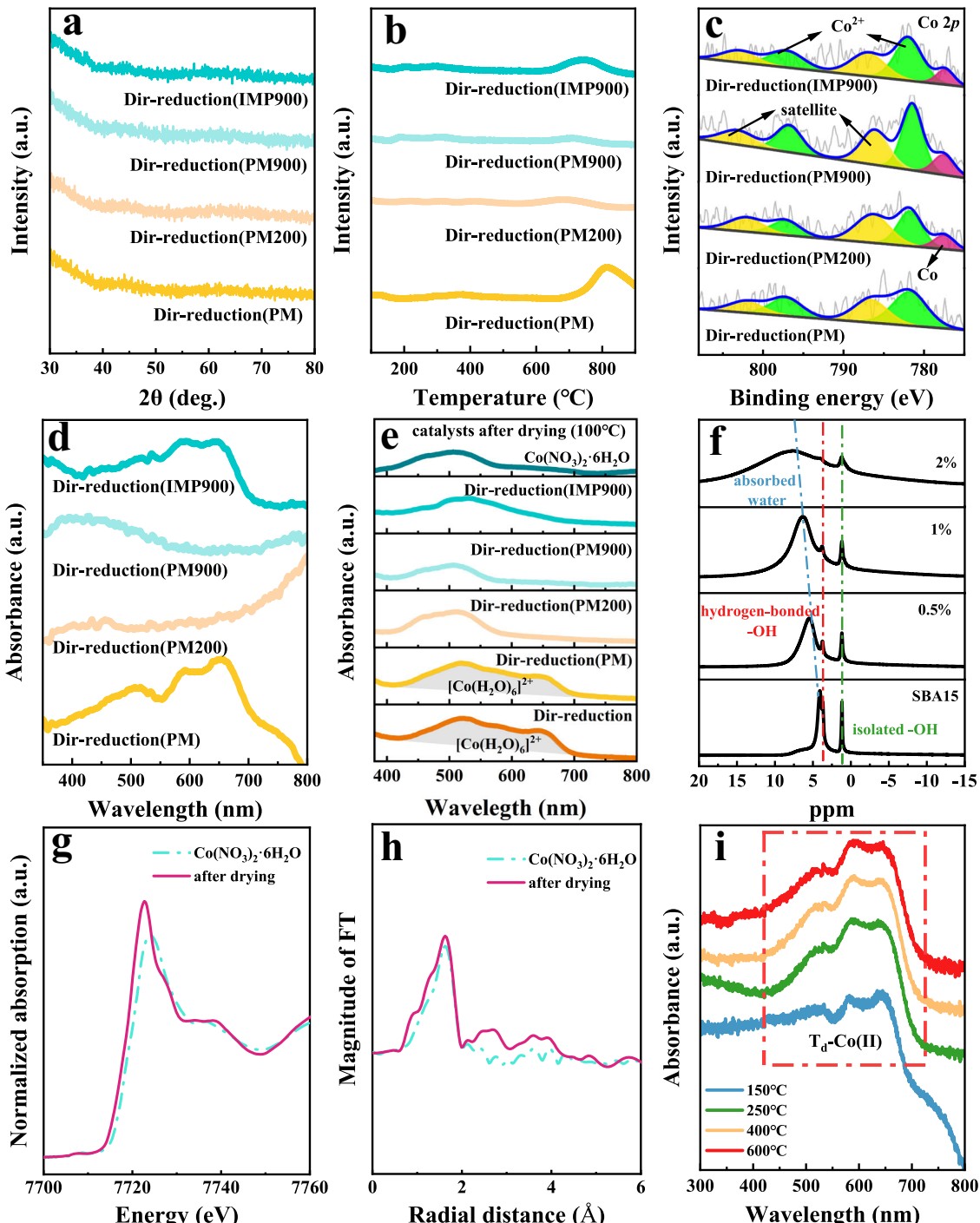

**Fig. 3 | Characterizations of verifying the effect of absorbed water on $T_d$-Co(II) formation. a** XRD pattens, **b** in-situ $H_2$-TPR profiles, **c** quasi in-situ Co 2$p$ XPS spectra, **d** ex-situ UV-vis spectra of the four designed catalysts; **e** UV-vis spectra of the dried catalyst precursors (without reduction treatment), **f** $^1$H-NMR of the dried catalyst precursors with different Co loadings; **g** the normalized intensity of Co K-edge XANES spectra, **h** the corresponding Fourier transformation of k$^3$-weighted EXAFS oscillation of the dried precursor for Dir-reduction catalyst; **i** UV-vis spectra of the Dir-reduction catalysts by using different reduction temperature.

Whereas, this peak was not found in the precursor of Dir-reduction (PM200) catalyst, since the absorbed water was completely removed. Inversely, one peak consistent with $Co(NO_3)_2 \cdot 6H_2O$ was appeared. Figure 3f presented the $^1$H-NMR of the dried precursor for Dir-reduction catalysts (after drying at 100 °C). Two resonances at around 1.2 ppm and 3.7 ppm were detected on the pure SBA-15 support. Normally, the peak at 3.7 ppm was assigned to the hydrogen-bonded OH[42], while the sharp peak at 1.2 ppm was assigned to the isolated OH[43]. Besides, a peak that emerged at around 4.9 ppm was related to

the adsorbed water on the SBA-15 surface[44]. Notably, with the increase of Co loading, the peak of adsorbed water was widened and shifted to the higher value position, which, together with the results of UV-vis from Fig. 3e, can further illustrate that this part of absorbed water was existed in the form of $[Co(H_2O)_6]^{2+}$. Namely, the structure of $[Co(H_2O)_6]^{2+}$ was well maintained after drying at 100 °C. As shown in Fig. 3g, the Co K-edge XANES spectra of the dried precursor of Dir-reduction catalyst was similar to that of $Co(NO_3)_2 \cdot 6H_2O$, but the changes in the electronic environment of Co was evidenced by the

slight shift of absorption edge. In Fig. 3h, the Fourier transformation of the $k^3$-weighted EXAFS showed that Co-O bonds within the CN of both Co-O1 and Co-O2 around 6 were observed, where their bond lengths were 2.089 Å and 3.514 Å, respectively, which displayed slight difference when comapred with the referenced sample of $Co(NO_3)_2 \cdot 6H_2O$. As a result, it means that the cobalt species were no longer in the form of cobalt nitrate at this time. The above experimental results lead to the conclusion that the existence of sufficient adsorbed water induced the transformation of precursor of $Co(NO_3)_2$ into intermediate of $[Co(H_2O)_6]^{2+}$, and finally, $T_d$-Co(II) was gained after $H_2$-direct reduction.

Following, we carried out UV-vis characterization of the catalysts precursors reduced directly at different temperatures (Fig. 3i). Interestingly, it is found that the characteristic peaks of $T_d$-Co(II) appeared at 150 °C, and this structure was well preserved at higher reduction temperature of 250 °C, 400 °C, and 600 °C. It is reported that dehydroxylation of $SiO_2$ started at least higher than 190 °C[45], and from this point of view, it is a concern in our study to reveal how the Co(II) replace hydroxyl group to form the stable $T_d$-Co(II) structure when dehydration condensation was performed below 190 °C. We hypothesized, based on the above results, the formation of $[Co(H_2O)_6]^{2+}$ within the presence of adsorbed water may promote the dehydration condensation of surface hydroxyl sites.

## Mediating mechanism of adsorbed water

In-situ DRIFTS of $NH_3$ adsorption-desorption and in-situ FT-IR were applied to reveal the changes of catalyst surface groups during $H_2$-direct reduction process, so as to illustrate the role of $[Co(H_2O)_6]^{2+}$ in the formation of $T_d$-Co(II). In view of the fact that $NH_3$ can interact with hydroxyl groups[46], which was regarded as a probe molecule to quantitatively analyze the concentration of surface OH on the SBA-15. If dehydration condensation can be facilitated by the presence of $[Co(H_2O)_6]^{2+}$, the residual amount of hydroxyl on Co/SBA-15 was supposed to be less than that of SBA-15. Consequently, in-situ DRIFTS of $NH_3$ adsorption-desorption was performed in Figs. 4e–3g. Obviously, the peak areas of both Co/SBA-15 and SBA-15 decreased with increasing reduction temperature, which was due to the gradual initiation of dehydroxylation during the heating process. More importantly, it is seen from Fig. 4e and Fig. 4f that the intensity of the peak over Co/SBA-15 was lower than that of SBA-15 at the same reduction temperature, and the peak area of Co/SBA-15 catalyst was smaller than that of the SBA-15. This difference was more distinct at high temperatures of 400 °C and 600 °C (Fig. 4g), which supported our suspicion that $[Co(H_2O)_6]^{2+}$ exactly accelerated the condensation of hydroxyl sites.

Meanwhile, in Fig. 4a, as for the Dir-reduction sample, split peaks of free-$NO_3^-$ at 1340 cm$^{-1}$ and 1410 cm$^{-1}$ were discovered[47]. When increasing temperature, the disappearance of these two split peaks was accompanied by the display of peak at 1530 cm$^{-1}$, illustrating that free-$NO_3^-$ was changed into the monodentate nitrate[48]. Notably, the nitrate was completely removed at high temperature of 600 °C, suggesting the transformation of cobalt nitrates into the stable Co-O-Si species[49]. Besides, the peak of adsorbed water at 1630 cm$^{-1}$ also disappeared gradually with the increase of temperature[50], indicating the consumption and transformation of adsorbed water. As for the Dir-reduction (PM) catalyst (Fig. 4b), it showed similar peak pattern to that of the Dir-reduction catalyst (Fig. 4a), illustrating that the adsorbed water substituted $NO_3^-$ in the precursor of $Co(NO_3)_2$ to form hydrate of $[Co(H_2O)_6]^{2+}$, resulting in the generation of free-$NO_3^-$. The difference is that the intensity of the diffraction peak in Fig. 4c was lower than that in Fig. 4b, which was caused by the fact that lower amount of adsorbed water in the Dir-reduction (IMP900) catalyst was not sufficient to obtain a higher amount of $[Co(H_2O)_6]^{2+}$. By contrast, similar processes were not observed on the Dir-reduction (PM900) catalyst (Fig. 4d). In particular, only one peak at 1512 cm$^{-1}$ attributed to $Co(NO_3)_2 \cdot 6H_2O$ was emerged[49], which is in agreement with the result of the UV-vis from

Fig. 3e. In conclusion, it is precisely because the presence of absorbed water which was in favor of the formation of $[Co(H_2O)_6]^{2+}$ made Co species easier to be transferred into $T_d$-Co(II).

DFT calculations were carried out to investigate the importance of the intermediate of hydrated cobalt species. The catalyst was orderly treated by the impregnation and drying processes, then Co was existed in the form of $[Co(H_2O)_6]^{2+}$(the coordination number and bond distance were calculated from XANES data). Notably, electrostatic interaction between fully coordinated $[Co(H_2O)_6]^{2+}$ and the hydroxyls on the support was important in terms of stabilizing the precursors. Namely, $[Co(H_2O)_6]^{2+}$ was supposed to be connected by the hydrogen bond (electrostatic interaction) between the oxygen in the silicon hydroxyl group and the hydrogen in the water of the $[Co(H_2O)_6]^{2+}$ (Fig. 4h, Supplementary Fig. 18, Supplementary Fig. 19), then the calculated BE of −8.03 kcal/mol (−0.348 eV) from BE = E($[Co(H_2O)_6]^{2+}$-$SiO_2$)−E($SiO_2$)−E($[Co(H_2O)_6]^{2+}$) indicated that the $[Co(H_2O)_6]^{2+}$-$SiO_2$ formed by the hydrogen bond between $SiO_2$ and $[Co(H_2O)_6]^{2+}$ was stable in the catalyst.

Subsequently, free energy for dehydroxylation of $SiO_2$ and Co/$SiO_2$ were calculated, and the dehydroxylation process of $SiO_2$ was exhibited in Fig. 4h and Supplementary Fig. 20. Since the energy barrier (1.264 eV) from IS to TS was the highest among all steps, it was considered as the rate-determining step. With the dehydroxylation carried out, the hydroxyl was removed from silanol groups. The reaction between the surface silanol groups (condensation) lead to the formation of Si-O-Si bonds and molecular water (($\equiv$Si-OH) + ( $\equiv$Si-OH) → ($\equiv$Si-O-Si $\equiv$) + $H_2O$), as demonstrated in TS and FS from Fig. 4h and Supplementary Fig. 20, with the free energy of 0.449 eV and −0.796 eV, respectively. By contrast, the dehydroxylation process of Co/$SiO_2$ was compared to that of $SiO_2$ (Fig. 4h and Supplementary Fig. 19). Two processes were simulated over Co/$SiO_2$. The first process was $[Co(H_2O)_6]^{2+}$ binding to $SiO_2$ through hydrogen bond, and the second process was dehydroxylation. It is seen that $[Co(H_2O)_6]^{2+}$ was stabilized by Si-OH via hydrogen bond with a free energy of −0.291 eV, which illustrated that the transformation from $[Co(H_2O)_6]^{2+}$ and $SiO_2$ into $[Co(H_2O)_6]^{2+}$-$SiO_2$ was spontaneous. Following, the dehydroxylation stage was proceeded, and $[Co(H_2O)_6]^{2+}$ occupied the site of dehydroxylation, with a free energy of 0.069 eV. Then, Co-O-Si bonds were gradually formed at high temperature (TS, Fig. 4h). It is obviously that the rate-determining step in the dehydroxylation over Co/$SiO_2$ was from IM1 to TS, with the energy barrier of 0.634 eV (Fig. 4h), which was lower than that of $SiO_2$. After that, the free energy of IM2 and FS was 0.176 eV and −1.129 eV (Fig. 4h), respectively, suggesting that the Td-Co(II) was acquired spontaneously due to the decrease of free energy. As a result, DFT calculations from Fig. 4h suggested that the energy barrier for dehydroxylation over Co/$SiO_2$ was lower than that over $SiO_2$, confirming that the existence of $[Co(H_2O)_6]^{2+}$ promoted the dehydroxylation on the silica support.

As indicated in previous studies, the dehydroxylation temperature on silica started from 190 °C, while the decomposition temperature of cobalt nitrate was about 240 °C[51]. As a result, in such a close range of temperature, it was difficult for Co(II) to interact with oxygen in the hydroxyl group to form a stable Co-O-Si structure. Inversely, Co oxides were more readily available. Surprisingly, the formation of $[Co(H_2O)_6]^{2+}$ reduced the energy barrier of the condensation process on the silica, which made it easier for Co(II) to be immobilized by the hydroxyl groups. This result well interpreted why the UV-Vis spectra in Fig. 3i found the characteristic peak of $T_d$-Co(II) at low temperature of 150 °C, and also in accord with the result of $NH_3$-DRIFTS in Fig. 4e−g that the presence of $[Co(H_2O)_6]^{2+}$ promoted the condensation of the higher amount of hydroxyl groups.

In conclusion, it is evidenced that the cobalt was in the form of $[Co(H_2O)_6]^{2+}$ over Dir-reduction and Dir-reduction(PM) catalysts, which contained abundant absorbed water. In contrast, cobalt nitrate was presented over these Dir-reduction(PM200) and Dir-

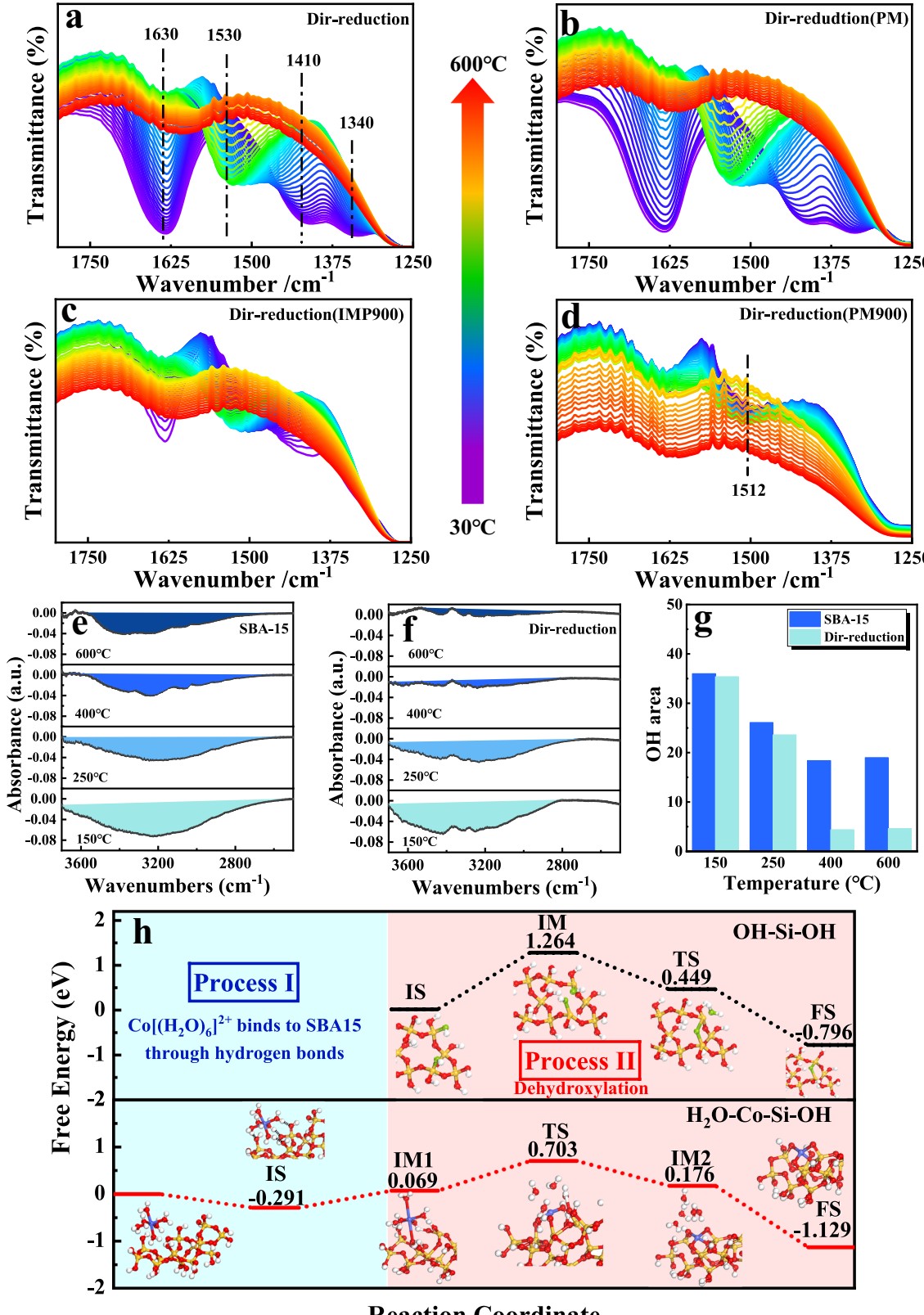

**Fig. 4 | Analysis of mediating mechanism of adsorbed water.** In-situ FT-IR spectra of **a** Dir-reduction, **b** Dir-reduction (PM), **c** Dir-reduction (IMP900), **d** Dir-reduction (PM900) during $H_2$-direct reduction process; In-situ $NH_3$-DRIFS of **e** SBA-15 and **f** Dir-reduction catalyst in the heating process, **g** peak area of hydroxyl group over SBA-15 and Dir-reduction catalyst from $NH_3$-DRIFS; **h** DFT calculations of the free energy of dehydroxylation.

reduction(PM900) samples, which were lacking in absorbed water. The above results indicated that cobalt nitrate can be converted into $[Co(H_2O)_6]^{2+}$ in the presence of adsorbed water, and $[Co(H_2O)_6]^{2+}$ made the highly dispersed $T_d$-cobalt(II) sites to be available during direct $H_2$-reduction process. Subsequently, a systematic in-situ characterizations suggested that the presence of $[Co(H_2O)_6]^{2+}$ promoted the process of dehydroxylation, and the following DFT calculation also proved the importance of the intermediate of $[Co(H_2O)_6]^{2+}$. Besides, as for the specifical role of direct $H_2$-reduction process, it was intended as a way to avoid the oxidation of Co species and facilitate the acquisition of tetrahedral cobalt(II). When adsorbed water was absent, there was no way to obtain highly dispersed $T_d$-Co(II). Hence, the necessary condition for the availability of highly dispersed $T_d$-cobalt(II) sites was the presence of adsorbed water, rather than the direct $H_2$-reduction treatment.

## Mechanism of propane dehydrogenation

Figure 5a displayed the reactivity of $H_2$-reduction and Dir-reduction catalysts with Co content of 2% toward PDH at 600 °C. A high $C_3H_8$ conversion (37%) and $C_3H_6$ selectivity (96%) were achieved over the Dir-reduction catalyst. In contrast, the conversion and selectivity over the $H_2$-reduction catalyst were 15 and 92%, respectively. More importantly, when comparing Dir-reduction sample with other reported cobalt-based catalysts[27,32,34,35,52–54], it performed apparently the maximum reaction rate (Fig. 5b), suggesting the developed catalyst gives a potential application for PDH. Meanwhile, PDH reactivity comparing Dir-reduction sample with other reported cobalt-based catalysts[27,32,34,35,52–54], it performed apparently the maximum reaction rate (Fig. 5b), suggesting the developed catalyst gives a potential application for PDH. Meanwhile, PDH reactivity of the Dir-reduction catalysts with Co loading of 4 and 6% was recorded in Supplementary Fig. 21. Similarly, the performance of the Dir-reduction catalysts was better than that of the $H_2$-reduction catalysts. Besides, the reactivity results of the designed samples from Fig. 2 were recorded in Supplementary Fig. 22, and $C_3H_8$ conversion was ranked as follows: Dir-reduction > Dir-reduction (PM) > Dir-reduction (IMP900) > Dir-reduction (PM900), which was consistent with our expectations that the content of highly dispersed and stable $T_d$-Co(II) was correlated positively with the PDH reactivity. It is known that the structure of $Co_3O_4$ and metallic Co were most likely led to the cracking of C-C bonds, causing the formation of $CH_4$ and some coke precursors[34,55], while highly dispersed $T_d$-Co(II) were supposed to be efficient in C-H activation for hydrocarbons[52]. This clearly explained why the Dir-reduction catalyst performed better catalytic reactivity for the PDH.

In-situ FT-IR was developed to further investigate the difference in the activation of propane and the desorption of propene between Dir-reduction and $H_2$-reduction catalysts. In the $C_3H_8$ adsorption FT-IR spectra (Figs. 5c, 4d), the bands at 2967 cm$^{-1}$, 1460 cm$^{-1}$ and 1370 cm$^{-1}$ assigned to $C_3H_8$ were detected[20]. It is seen from Fig. 5c that new bands at 1626 cm$^{-1}$, 1875 cm$^{-1}$, and 2004 cm$^{-1}$ representing for the formation of $C_3H_6$ were emerged when the Dir-reduction catalyst was heated to 300 °C[54], suggesting $C_3H_8$ was participated in the reaction to produce $C_3H_6$ over the catalyst, while no new peak was detected on the $H_2$-reduction catalyst, which illustrated the better $C_3H_8$ activation reactivity over the Dir-reduction sample. Furthermore, $C_3H_6$ adsorption FT-IR measurements were given in Fig. 5e, f. Three peaks located at 1626 cm$^{-1}$, 1875 cm$^{-1}$ and 2004 cm$^{-1}$ were also found on the Dir-reduction catalyst (Fig. 5e), which was attributed to the characteristic adsorption bands of $C_3H_6$. By contrast, the peak located at 1626 cm$^{-1}$, assigned to the adsorbed $C_3H_6$ on the $T_d$-Co(II), can hardly be observed on the $H_2$-reduction catalyst (Fig. 5f), suggesting strong interaction between the adsorbed $C_3H_6$ and the $H_2$-reduction catalyst. It is noted from Fig. 5f that after desorption at 300 °C, a broad peak observed from 1530 cm$^{-1}$ to 1670 cm$^{-1}$ resulted from the formed carbonaceous products[28], and the peak intensity increased with the desorption time,

which indicated that $C_3H_6$ was interacted strongly with the $H_2$-reduction catalyst, thus causing the formation of carbonaceous species. Subsequently, in-situ FT-IR of propene hydrogenation process (the reverse of dehydrogenation) was performed in Supplementary Fig. 23. It is observed that the peak at 1626 cm$^{-1}$ attributed to the adsorbed $C_3H_6$ on the $T_d$-Co(II) was absent the $H_2$-reduction catalyst. Moreover, carbonaceous species disappeared on $H_2$-reduction catalyst during the propylene hydrogenation process. When combining the results of Supplementary Fig. 23 and Fig. 5f, it is implied that $C_3H_6$ was easily cleaved to carbonaceous species, which caused the poor selectivity of the $H_2$-reduction catalyst. Following, density functional theory (DFT) calculations were applied to illustrate the mechanism of the PDH reactivity on the Dir-reduction and $H_2$-reduction catalysts, and the geometries of the transition state (TS) were exhibited in Supplementary Fig. 24 and Supplementary Fig. 25. It is obtained from Fig. 5g that energy barriers for the activation of first and second C-H bonds on the Dir-reduction catalyst were 0.40 eV and 0.33 eV, respectively, while $H_2$-reduction catalyst possessed higher energy barrier of 0.52 eV and 1.02 eV for the reaction. Regarding the above results, our calculations are consistent with the experimental results of a higher PDH reactivity for the Dir-reduction Co/SBA-15 catalyst.

Finally, a series of characterization tests were also performed to investigate whether $T_d$-Co(II) can be stably presented during the reaction. UV-vis and $H_2$-TPR of the spent Dir-reduction catalyst during the first 30 min of the reaction were tested (Supplementary Fig. 26). It is obvious that the structure of $T_d$-Co(II) remained intact during the reaction. Moreover, XPS, UV-vis and $H_2$-TPR of the regenerated Dir-reduction catalyst from Supplementary Figs. 27 and 28 illustrated that $T_d$-Co(II) was not oxidized by air after regeneration at 600 °C. In conclusion, highly dispersed tetrahedral Co(II) sites obtained from Dir-reduction catalyst shows the high ability to break C-H and maintains a well-defined structure during the reaction.

In summary, a new understanding of adsorbed water on the formation of highly dispersed $T_d$-cobalt(II) sites from direct $H_2$-reduction process was illustrated. It is indicated the cobalt interacted with the adsorbed water that were attached to the hydroxyl group on the silica support through hydrogen bond to form $[Co(H_2O)_6]^{2+}$. Subsequently, the formed $[Co(H_2O)_6]^{2+}$ was anchored by the oxygen in the hydroxyl group to form the highly dispersed $T_d$-cobalt(II) sites. Significantly, the presence of $[Co(H_2O)_6]^{2+}$ reduced the free energy of dehydroxylation process over the silica support of SBA-15, which created a beneficial condition for the formation of required $T_d$-Co(II) sites. The obtained Co/SBA-15 catalyst from direct $H_2$-reduction process displayed better reactivity than the reported cobalt-based catalysts toward PDH reaction. As a result, the present work provides a simple and effective approach to designing high reactivity of cobalt-based PDH catalysts.

## Methods

### Catalyst synthesis

The catalysts of Co/SBA-15 were prepared by incipient wetness impregnation method, using $Co(NO_3)_2\cdot6H_2O$ as the precursor. The nominal amount of cobalt in the prepared catalysts was 2, 4, 6wt.%. Typically, the desired weight of $Co(NO_3)_2\cdot6H_2O$ was dissolved in 8 mL of deionized water, 2 g of SBA-15 was added with continuous stirring. Subsequently, the obtained mixture was dried at 100 °C overnight. The drying sample was performed directly in 10% $H_2$/Ar flow at 600 °C for 2 h (Dir-reduction catalyst). Meanwhile, the drying mixture was calcined in air at 600 °C for 5 h to achieve Air-calcination sample, and then was reduced in 10% $H_2$/Ar flow at 600 °C for 2 h to obtain an $H_2$-reduction catalyst.

### Catalysts characterization and DFT calculations

Catalysts characterization, which includes powder X-ray diffraction (XRD), quasi in-situ X-ray photoelectron spectroscopy (XPS), $H_2$-temperature-programmed reduction ($H_2$-TPR), Ultraviolet-visible (UV-vis),

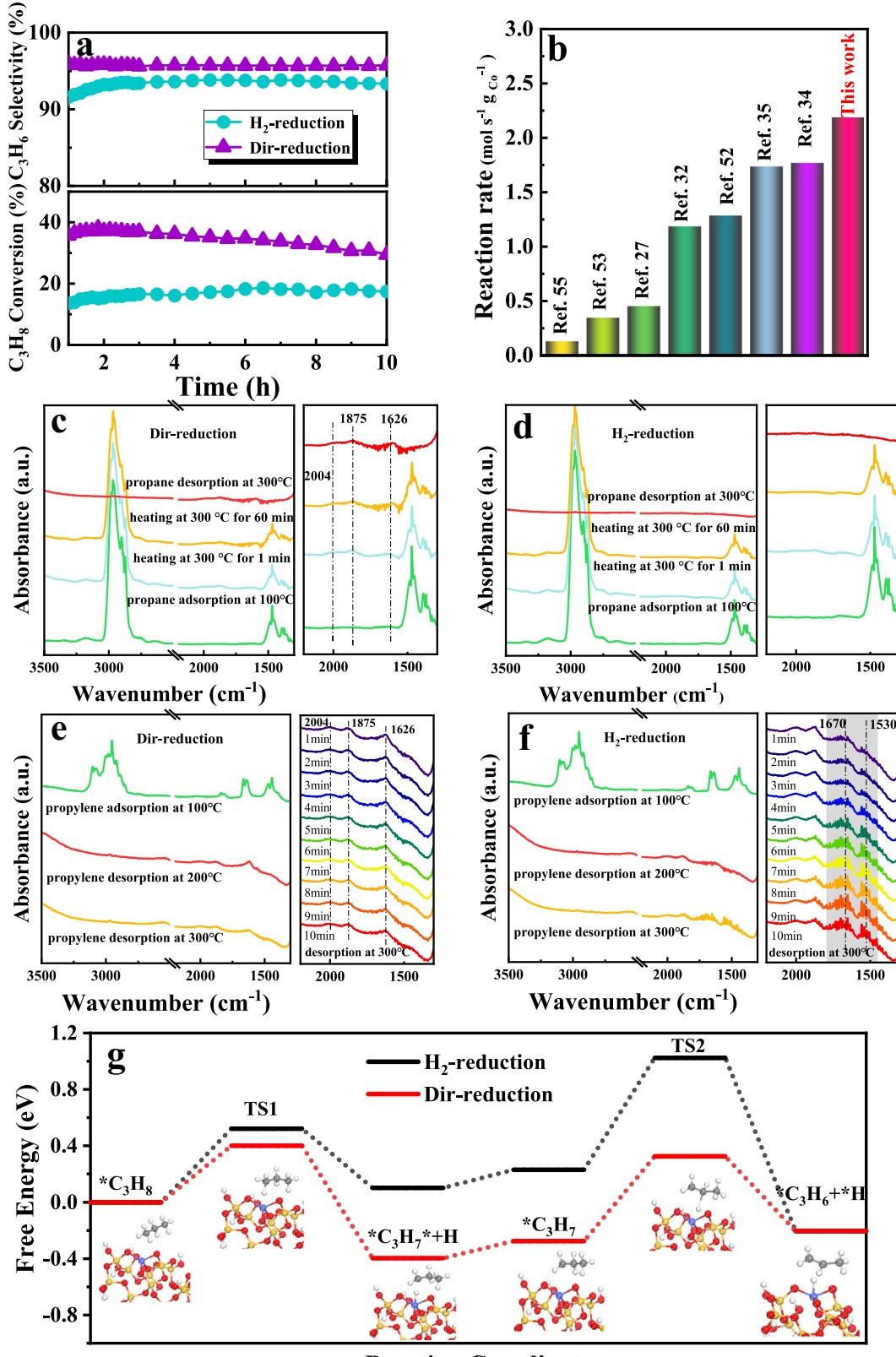

**Fig. 5 | Mechanism of propane dehydrogenation. a** C₃H₈ conversion, C₃H₆ selectivity, **b** comparative study of reaction rate for the reported cobalt-based catalysts; C₃H₈ adsorption in-situ FT-IR of **c** Dir-reduction catalyst and **d** H₂-reduction catalyst; C₃H₆ adsorption in-situ FT-IR of **e** Dir-reduction catalyst and (**f**) H₂-reduction catalyst; **g** Free energy diagrams for PDH.

Transmission electron microscopy (TEM), $^1H$ NMR, in situ $NH_3$-DRIFTS, in situ FT-IR and DFT computation were given in the Supporting Information (SI).

## Catalytic tests

The catalytic reactivity of PDH was evaluated in a fixed-bed reactor under atmospheric pressure. In a typical test, 0.4 g of catalyst (sieved between 40 and 60 mesh) was loaded in the center of the reactor. Then, the reaction gas of $C_3H_8$ (5 mL/min) and $N_2$ (30 mL/min) was introduced for reactivity evaluation. The reaction was carried out at 600 °C, and the feed and products were analyzed by an online gas chromatograph (Fuli 9790GC) equipped with TCD and FID detectors. The conversion of $C_3H_8$ and selectivity of $C_3H_6$ were calculated as follows:

$$C_3H_8\,conversion(\%) = \frac{C_3H_{8in} - C_3H_{8out}}{C_3H_{8in}} \times 100 \qquad (1)$$

$$C_3H_6\,selectivity(\%) = \frac{nC_3H_6}{nC_3H_6 + (2/3)nC_2H_6 + (2/3)nnC_2H_4 + (1/3)nCH_4} \qquad (2)$$

where n is the number of moles of hydrocarbons.

## Data availability

The data that support the findings of this study are available from the corresponding author upon request.

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

## Acknowledgements

The authors wish to acknowledge the financial support provided by the National Natural Science Foundation of China (21966018, 22006059, 21968015, 42030712), Excellent Youth Project of Natural Science Foundation of Yunnan Province (202201AW070007), Young Academic and Technical Leader Raising Foundation of Yunnan Province (202205AC160011), Key Project of Natural Science Foundation of Yunnan Province (No. 202101AS070026).

## Author contributions

Y.L. conceived the research. D.H. conceived the research. Z.H. performed the catalyst synthesis and characterizations. W.D. performed the catalytic tests works and co-wrote the manuscript in the discussion. K.L. carried out the XAFS analysis. G.J. carried out the XAFS analysis. All authors discussed the results and commented on the manuscript.

## Competing interests

The authors declare no competing interests.
