## [Peer Review File · Nature Communications]

Illustrating new understanding of adsorbed water on silica for inducing tetrahedral cobalt(II) for propane dehydrogenationREVIEWER COMMENTS

Reviewer #1 (Remarks to the Author):

I read the manuscript with great interests. Authors described the role of water during the synthesis of Co-based catalyst. The results showed the the hydrates is critical to the tetrahedral Co site formation. The as-synthesized catalyst was attempted to the PDH and results were encouraging. The topic is of great importance as the synthesis of catalyst is utmost crucial to determine the catalytic activity. Any details such as the role of water regrading the understanding on catalyst synthesis should draw attention to the community. Authors attempted to show the results of water role, nonetheless, the discussion seems poor, for instance, how the fully coordinated $\text{Co}(\text{H}_2\text{O})_6$ can coordinate with hydroxyls without the consideration of electrostatic interaction which I think is more important in terms of stabilizing the precursors. In other words, the discussion is not fully convincing at this moment, the calculation is also not that detailed to understand. All the figures should be of high resolution.

Reviewer #3 (Remarks to the Author):

In this paper, the authors prepared the catalyst with highly dispersed Td-cobalt(II) on the surface of silica, achieving high catalytic performance of propane dehydrogenation by inducing adsorbed water. In addition, a systematic characterization and DFT calculation have proved the existence of the adsorbed water and the importance of the intermediate of $[\text{Co}(\text{H}_2\text{O})_6]^{2+}$, respectively. Overall, this work provides a novel view to understand the function of adsorbed water on SBA-15 and have the qualification for the publication on Nature Communications, if the authors adequately take my suggestions into consider:

1. In the line 172, the authors claimed that “in situ XPS measurement from Fig. S5 confirmed again that only Co(II) species were found on the Dir-reduction sample.” However, Fig. S5 is not a in situ X-ray photoelectron spectroscopy. How to define this figure need to reconsider.
2. In the line 184, the authors mentioned that “... Co was connected strongly within the silica support and hard to be reduced.” However, it is better to use the word of “Co species” to replace the word of “Co” if the authors do not want to mention the valence state of cobalt component, because the latter will lead readers to consider the Co species are metallic Co.
3. According to the result of H₂-TPR (Fig. 1c), the authors said “... all of the Co oxides on the surface were reduced by H₂.” Therefore, the authors may need to explain that the high content of Co²⁺ of the H₂-reduction catalyst in Fig. 1b.
4. The notes of Fig. 1c, “ $\text{Co}_3\text{O}_4 \rightarrow \text{CoO} \rightarrow \text{Co}$ ” and “Co²⁺”, should be unified.
5. In the line 231, the authors said “then mixed physically with $\text{Co}(\text{NO}_3)_2 \cdot 6\text{H}_2\text{O}$ in glove box before conducting H₂-direct reduction (Dir-reduction (PM200))”. In the line 235 the authors said “Afterwards, 900oC-calcined SBA-15 was mixed physically with $\text{Co}(\text{NO}_3)_2 \cdot 6\text{H}_2\text{O}$ to obtain Dir-reduction (PM900) sample”. These ambiguous sentences would confuse audiences whether the PM200 was reduced in H₂-direct reduction process or not.
6. In the lines 257-261, the authors shown the H₂-TPR curves of PM200, PM900 and IMP900 catalyst, which revealed that Co_3O_4 is in the catalysts. Thus, XPS spectra of these catalysts should have the peak

of Co^{3+} , while it is absent in Fig. 2c.

7. The reduction temperature of contrast samples (PM200, PM900 and IMI900) should be presented. The Co/SBA-15 is reduced 600°C (line 538), if the contrast samples were also reduced at 600°C, the authors should explain why in the H₂-TPR profiles (Fig. 2b) reduction peaks occur below 600°C.

8. The authors should explicit the function of the H₂ reduction process, because the authors say H₂ reduction process is to generate highly dispersed Td-cobalt(II) sites in the abstract, while the authors also say that this process is to induce adsorb water in the line 106. Thus, the authors may need to explain it.

9. In addition, it is better to enlarge the font size of some icons, like Fig. 1(a, b, c, d, e, f).

10. Please specify the manufacturers and purity of all the chemicals and materials used in this work.

REVIEWER COMMENTS

Reviewer #1 (Remarks to the Author):

I read the manuscript with great interests. Authors described the role of water during the synthesis of Co-based catalyst. The results showed the hydrates is critical to the tetrahedral Co site formation. The as-synthesized catalyst was attempted to the PDH and results were encouraging. The topic is of great importance as the synthesis of catalyst is utmost crucial to determine the catalytic activity. Any details such as the role of water regarding the understanding on catalyst synthesis should draw attention to the community. Authors attempted to show the results of water role, nonetheless, the discussion seems poor, for instance, how the fully coordinated $\text{Co}(\text{H}_2\text{O})_6$ can coordinate with hydroxyls without the consideration of electrostatic interaction which I think is more important in terms of stabilizing the precursors. In other words, the discussion is not fully convincing at this moment, the calculation is also not that detailed to understand. All the figures should be of high resolution.

Reviewer #3 (Remarks to the Author):

In this paper, the authors prepared the catalyst with highly dispersed Td-cobalt(II) on the surface of silica, achieving high catalytic performance of propane dehydrogenation by inducing adsorbed water. In addition, a systematic characterization and DFT calculation have proved the existence of the adsorbed water and the importance of the intermediate of $[\text{Co}(\text{H}_2\text{O})_6]^{2+}$, respectively. Overall, this work provides a novel view to understand the function of adsorbed water on SBA-15 and have the qualification for the publication on Nature Communications, if the authors adequately take my suggestions into consider:

1. In the line 172, the authors claimed that “*in situ* XPS measurement from Fig. S5 confirmed again that only Co(II) species were found on the Dir-reduction sample.”

However, Fig. S5 is not a *in situ* X-ray photoelectron spectroscopy. How to define this figure need to reconsider.

2. In the line 184, the authors mentioned that "... Co was connected strongly within the silica support and hard to be reduced." However, it is better to use the word of "Co species" to replace the word of "Co" if the authors do not want to mention the valence state of cobalt component, because the latter will lead readers to consider the Co species are metallic Co.

3. According to the result of H₂-TPR (Fig. 1c), the authors said "... all of the Co oxides on the surface were reduced by H₂." Therefore, the authors may need to explain that the high content of Co²⁺ of the H₂-reduction catalyst in Fig. 1b.

4. The notes of Fig. 1c, "Co₃O₄→CoO→Co" and "Co²⁺", should be unified.

5. In the line 231, the authors said "then mixed physically with Co(NO₃)₂·6H₂O in glove box before conducting H₂-direct reduction (Dir-reduction (PM200))". In the line 235 the authors said "Afterwards, 900 °C-calcined SBA-15 was mixed physically with Co(NO₃)₂·6H₂O to obtain Dir-reduction (PM900) sample". These ambiguous sentences would confuse audiences whether the PM200 was reduced in H₂-direct reduction process or not.

6. In the lines 257-261, the authors shown the H₂-TPR curves of PM200, PM900 and IMP900catalyst, which revealed that Co₃O₄ is in the catalysts. Thus, XPS spectra of these catalysts should have the peak of Co³⁺, while it is absent in Fig. 2c.

7. The reduction temperature of contrast samples (PM200, PM900 and IMI900) should be presented. The Co/SBA-15 is reduced 600 °C (line 538), if the contrast samples were also reduced at 600 °C, the authors should explain why in the H₂-TPR profiles (Fig. 2b) reduction peaks occur below 600 °C.

8. The authors should explicit the function of the H₂ reduction process, because the authors say H₂ reduction process is to generate highly dispersed Td-cobalt(II) sites in

the abstract, while the authors also say that this process is to induce adsorb water in the line 106. Thus, the authors may need to explain it.

9. In addition, it is better to enlarge the font size of some icons, like Fig. 1(a, b, c, d, e, f).

10. Please specify the manufacturers and purity of all the chemicals and materials used in this work.

Reviewer #1

Comment (1): Any details such as the role of water regarding the understanding on catalyst synthesis should draw attention to the community. Authors attempted to show the results of water role, nonetheless, the discussion seems poor, for instance, how the fully coordinated $\text{Co}(\text{H}_2\text{O})_6$ can coordinate with hydroxyls without the consideration of electrostatic interaction which I think is more important in terms of stabilizing the precursors. In other words, the discussion is not fully convincing at this moment, the calculation is also not that detailed to understand.

Reply: We feel great thanks for your professional review work on our article. As you are concerned, there are several problems that need to be addressed. According to your nice suggestions, we have made extensive corrections to our previous draft, the detailed corrections are listed below.

1. First of all, we agree with the reviewer's opinion that the fully coordinated $[\text{Co}(\text{H}_2\text{O})_6]^{2+}$ was not directly coordinated with the hydroxyl group, instead, it was supposed to be connected by the hydrogen bond (electrostatic interaction) between the oxygen in the silanol groups and the hydrogen in the water of the $[\text{Co}(\text{H}_2\text{O})_6]^{2+}$, as shown in Supplementary Fig. 18.

Supplementary Fig. 18. The optimized structure of $[\text{Co}(\text{H}_2\text{O})_6]^{2+}$, $[\text{Co}(\text{H}_2\text{O})_6]^{2+}\text{-SiO}_2$ and SiO_2 .

In order to confirm that hydrogen bond stabilized the existence of $[\text{Co}(\text{H}_2\text{O})_6]^{2+}$. Following, the reactions between $[\text{Co}(\text{H}_2\text{O})_6]^{2+}$ and SiO_2 were studied via DFT calculations (the structures were presented in Supplementary Fig. 18). The binding energy (BE) was used to elucidate the interaction between $[\text{Co}(\text{H}_2\text{O})_6]^{2+}$ and SiO_2 , which was defined as:

$$\text{BE} = E([\text{Co}(\text{H}_2\text{O})_6]^{2+}\text{-SiO}_2) - E(\text{SiO}_2) - E([\text{Co}(\text{H}_2\text{O})_6]^{2+}) \quad (1)$$

The binding energies of $[\text{Co}(\text{H}_2\text{O})_6]^{2+}$, $[\text{Co}(\text{H}_2\text{O})_6]^{2+}\text{-SiO}_2$ and SiO_2 were respectively computed, and the value of the BE was -8.03 kcal/mol (-0.348 eV) after calculated by Eq 1. The value of the BE indicated that the $[\text{Co}(\text{H}_2\text{O})_6]^{2+}\text{-SiO}_2$ formed by hydrogen bond between SiO_2 and $[\text{Co}(\text{H}_2\text{O})_6]^{2+}$ was stable in the catalyst.

Subsequently, free energy for dehydroxylation of SiO_2 and Co/SiO_2 were calculated, and the dehydroxylation process of SiO_2 was exhibited in Fig. 3h and Supplementary Fig. 20. Since the energy barrier (1.264 eV) from IS to TS was the highest among all steps, it was considered as the rate-determining step. With the dehydroxylation carried out, the hydroxyl was removed from silanol groups. The reaction between the surface silanol groups (condensation) lead to the formation of Si-O-Si bonds and molecular water ($(\equiv\text{Si-OH}) + (\equiv\text{Si-OH}) \rightarrow (\equiv\text{Si-O-Si}\equiv) + \text{H}_2\text{O}$), as demonstrated in TS and FS from Fig. 3h and Supplementary Fig. 20, with the free energy of 0.449 eV and -0.796 eV, respectively. By contrast, the dehydroxylation process of Co/SiO_2 was compared to that of SiO_2 (Fig. 3h and Supplementary Fig. 19). Two processes were simulated over Co/SiO_2 . The first process was $[\text{Co}(\text{H}_2\text{O})_6]^{2+}$ binding to SiO_2 through hydrogen bond, and the second process was dehydroxylation. It is seen that

Fig. 3 (h). DFT calculations of the free energy for dehydroxylation.

$[\text{Co}(\text{H}_2\text{O})_6]^{2+}$ was stabilized by Si-OH via hydrogen bond with a free energy of -0.291 eV, which illustrated that the transformation from $[\text{Co}(\text{H}_2\text{O})_6]^{2+}$ and SiO_2 into $[\text{Co}(\text{H}_2\text{O})_6]^{2+}\text{-SiO}_2$ was spontaneous. Following, dehydroxylation stage was proceeded, $[\text{Co}(\text{H}_2\text{O})_6]^{2+}$ occupied the site of dehydroxylation, with a free energy of 0.069 eV. Then, Co-O-Si bonds were gradually formed at high temperature (TS, Fig. 3h). It is obviously that the rate-determining step in the dehydroxylation over Co/ SiO_2 was from IM1 to TS, with the energy barrier of 0.634 eV (Fig. 3h), which was lower than that of SiO_2 . After that, the free energy of IM2 and FS was 0.176 eV and -1.129 eV (Fig. 3h), respectively, suggesting that the Td-Co(II) was acquired spontaneously due to the decrease of free energy. As a result, DFT calculations from Fig. 3h suggested that the energy barrier for dehydroxylation over Co/ SiO_2 was lower than that over SiO_2 , confirming that the existence of $[\text{Co}(\text{H}_2\text{O})_6]^{2+}$ promoted the

dehydroxylation on the silica support.

We have added a detailed discussion for the DFT calculation in the revised Manuscript (line 410-495, page 18-22), by considering that electrostatic interaction between fully coordinated $[\text{Co}(\text{H}_2\text{O})_6]^{2+}$ and the hydroxyls on the support was important in terms of stabilizing the precursors.

Supplementary Fig. 19. The flow diagram for the dehydroxylation of hydroxyl groups on Dir-reduction catalyst.

Supplementary Fig. 20. The flow diagram for the dehydroxylation of hydroxyl groups on the pure SBA-15 catalyst (To make it easier to show the change of silanol groups, the oxygen was replaced with green atom).

2. Next, we have revised the whole discussions to improve the readability and understandability of the manuscript. In brief, it is evidenced that the cobalt was in the form of $[\text{Co}(\text{H}_2\text{O})_6]^{2+}$ over Dir-reduction and Dir-reduction(PM) catalysts, which contained abundant adsorbed water. In contrast, cobalt nitrate was presented over these Dir-reduction(PM200) and Dir-reduction(PM900) samples, which were lacking in adsorbed water. The above results indicated that cobalt nitrate can be converted into $[\text{Co}(\text{H}_2\text{O})_6]^{2+}$ in the presence of adsorbed water, and $[\text{Co}(\text{H}_2\text{O})_6]^{2+}$ made the highly dispersed Td-cobalt(II) sites to be available during direct H_2 -reduction process.

Subsequently, a systematic *in-situ* characterizations suggested that the presence of $[\text{Co}(\text{H}_2\text{O})_6]^{2+}$ promoted the process of dehydroxylation, and the following DFT calculation also proved the importance of the intermediate of $[\text{Co}(\text{H}_2\text{O})_6]^{2+}$.

Detailed revisions for the discussion can be found in the revised Manuscript (line 270-271, line 276-279, page 12; line 324-331, page 14-15; line 356-361, page 16; line 400-403, page 18; line 507-516, page 23).

Comment (2): All the figures should be of high resolution.

Reply: Thanks for your suggestions. We have replaced all the figures with high resolution in the revised Manuscript.

Reviewer #3

Comment (1): In the line 172, the authors claimed that “*in situ* XPS measurement from Fig. S5 confirmed again that only Co(II) species were found on the Dir-reduction sample.” However, Fig. S5 is not a *in situ* X-ray photoelectron spectroscopy. How to define this figure need to reconsider.

Reply: Thanks for your comments. Indeed, the present XPS measurement is quasi *in-situ* XPS. Namely, quasi *in-situ* X-ray photoelectron spectroscopy (XPS) measurement was performed on a Thermo ESCALAB 250Xi spectrometer, equipped with a monochromatic Al K α radiation source ($h\nu = 1486.6$ eV). In the reaction chamber, the samples (catalyst precursor (after impregnation) and Air-calcination sample) were respectively reduced in 10% H_2/Ar at 600 °C for 2 h. After treatment, the sample was cooled down to room temperature, and then directly transferred to analytical chamber in a vacuum (UHV XPS) to avoid exposure to air.

The revised description and experimental details for the procedure of quasi *in-situ* XPS have been added into the revised Manuscript (line 181, page 8; Fig.

2c, page 13) and revised Supporting Information (line 41-48, page 2-3; Supplementary Fig. 5, page 9), respectively.

Comment(2): In the line 184, the authors mentioned that "... Co was connected strongly within the silica support and hard to be reduced." However, it is better to use the word of "Co species" to replace the word of "Co" if the authors do not want to mention the valence state of cobalt component, because the latter will lead readers to consider the Co species are metallic Co.

Reply: Thank you for pointing this out. According to your suggestions, the word of "Co" was replaced by the word of "Co species" in the revised Manuscript (line 193, page 9).

Comment(3): According to the result of H₂-TPR (Fig. 1c), the authors said "... all of the Co oxides on the surface were reduced by H₂." Therefore, the authors may need to explain that the high content of Co²⁺ of the H₂-reduction catalyst in Fig. 1b.

Reply: Thanks for your remarks. This phenomenon was mainly caused by the fact that Co oxides in the bulk phase cannot be totally reduced to Co⁰ and the different depths of the samples detected by XPS and TPR techniques.

In general, metal oxides loaded on support were difficult to be completely reduced to metals. Gabor A. Somorjai et al. found that Co/TiO₂ was reduced under 20%H₂/Ar at 450 °C, a certain amount of Co²⁺/Coⁿ⁺ species were also detected by AP-XPS (*J. Am. Chem. Soc.* 2014, 136, 6). Co/Y zeolite was pretreated in H₂ at 450 °C for 8 h and then applied for Fischer-Tropsch reaction (CO + H₂). In spite of the fact that the catalyst was continuously under reducing atmosphere, the spent catalyst still exhibited a mixture of cobalt oxides and metallic cobalt (*Nat. Catal.* 2018, 1, 787-793). Moreover, the evolution of Ru/TiO₂ catalysts reduced at different temperature was

investigated with X-ray absorption spectroscopy (XAS), which suggested a gradual improvement in the reduction degree of Ru oxide to metallic phase with increasing reduction temperature. However, complete reduction of Ru was not achieved even at 600 °C (*Nat. Commun.* 2020, 11, 3185). Besides, Ding Ma et al. used quasi *in-situ* XPS to observe the change in valence of Ni under 15% CH₄/H₂ mixture at 590 °C, the emerged peak of both Ni⁰ and Ni²⁺ were detected in the Ni 2p spectra (*Chem.* 2023, 9, 1-15). The result of *in-situ* XPS was also illustrated that NiO was partially reduced to metallic Ni under H₂ at 590 °C (*Nat. Commun.* 2019, 10, 5181). From this point of view, it is deduced that crystalline cobalt oxide in our catalyst were difficult to be totally reduced.

On the other hand, XPS and TPR technique are both method for characterizing the surface properties of the catalysts. In general, XPS can detect composition changes within 2-3 nm on the catalyst surface (*Corrosion Sci.* 2020, 174, 108837). In contrast, TPR is highly influenced by experimental conditions, such as reactant concentration, flow rate, temperature and reactor size (*J. Catal.* 1983, 83, 323-335). Thus, it is not easy to achieve the same test depth as the XPS. In other words, XPS can detect species on the subsurface, while TPR can only detect species on the external surface. From the point of view of measuring depth, XPS technique was more likely to detect oxides that were not reduced in the subsurface. Hence, the above difference can be used to explain why there was a visible peak of Co²⁺ in both XPS spectra (Fig. 1b) and quasi *in-situ* X-ray photoelectron spectra (Supplementary Fig. 5), while no reduction peak was observed in the TPR profiles (Fig. 1c).

We have revised the corresponding expression to avoid unnecessary misunderstanding in the revised manuscript (line 186-188, page 8).

Comment(4): The notes of Fig. 1c, “Co₃O₄→CoO→Co” and “Co²⁺”, should be

unified.

Reply: Thank you for the suggestion. The notes of “ $\text{Co}_3\text{O}_4 \rightarrow \text{CoO} \rightarrow \text{Co}$ ” and “ Co^{2+} ” has been unified in the revised Fig. 1c (page 7) and the revised Manuscript (line 186, page 8).

Fig. 1(c). H₂-TPR profiles for Dir-reduction, Air-calcination and H₂-reduction catalysts.

Comment(5): In the line 231, the authors said “then mixed physically with $\text{Co}(\text{NO}_3)_2 \cdot 6\text{H}_2\text{O}$ in glove box before conducting H₂-direct reduction (Dir-reduction (PM200))”. In the line 235 the authors said “Afterwards, 900 °C-calcined SBA-15 was mixed physically with $\text{Co}(\text{NO}_3)_2 \cdot 6\text{H}_2\text{O}$ to obtain Dir-reduction (PM900) sample”. These ambiguous sentences would confuse audiences whether the PM200 was reduced in H₂-direct reduction process or not.

Reply: Thank you for your valuable suggestions. As a matter of fact, the four samples

of Dir-reduction (PM), Dir-reduction (PM200), Dir-reduction (PM900), Dir-reduction (IMP900) were all reduced in H₂-direct reduction process (catalyst precursor was reduced directly by H₂).

Based on your comments, we have made the corrections to make the sentences clear in the revised Manuscript (line 232-233, page 10; line 247-248, page 11; line 260, line 265, line 268-269, page 12).

Comment(6): In the lines 257-261, the authors shown the H₂-TPR curves of PM200, PM900 and IMP900 catalyst, which revealed that Co₃O₄ is in the catalysts. Thus, XPS spectra of these catalysts should have the peak of Co³⁺, while it is absent in Fig. 2c.

Fig. 2. (b) H₂-TPR profiles, (c) Co 2p XPS spectra, (d) UV-vis spectra of the four designed catalysts;

Reply: We really appreciate the careful check on the manuscript. According to the results of H₂-TPR and UV-vis in Fig. 2b, d, it is shown that the Co³⁺ was present in

PM200, PM900 and IMP900 catalysts, so the XPS spectra of these catalysts should display the peak of Co^{3+} . After re-analyze the XPS data, the results were shown in the above figure (Fig. 2c). The peaks at 779.5 eV and 794.6 eV were attributed to the surface Co^{3+} species. The peak at 777.5 eV was attributed to the metallic Co. However, in view of the reviewer's comment 7, we have re-characterized the experiments of Fig. 2b and Fig. 2c (the previous experiments of H_2 -TPR and Co 2p XPS were not carried out under *in situ* reaction conditions, and the reduced samples were subjected to reoxidation). Notably, the revised Fig. 2b and Fig. 2c are recorded in Comment(7).

Comment(7): The reduction temperature of contrast samples (PM200, PM900 and IMI900) should be presented. The Co/SBA-15 is reduced 600 °C (line 538), if the contrast samples were also reduced at 600 °C, the authors should explain why in the H_2 -TPR profiles (Fig. 2b) reduction peaks occur below 600 °C.

Reply: Thanks for your remarks. Indeed, the four contrast samples of PM, PM200, PM900 and IMP900 were all conducted by H_2 -direct reduction treatment (catalyst precursor was reduced directly by H_2), and the reduction temperature was 600 °C. However, the previous experiments of H_2 -TPR and Co 2p XPS were not carried out under *in situ* reaction conditions, and the reduced samples were easily re-oxidized in the air. As a result, we speculated that the reduction peak below 600 °C in Fig. 2b from Comment(6) was most likely formed by the surface Co oxides on the reduced samples. In order to eliminate the re-oxidation in the *ex situ* characterization, quasi *in-situ* XPS and *in-situ* TPR were employed, and the revised Fig. 2b and Fig. 2c are indicated as below.

In the revised Fig. 2b (*in-situ* TPR), Dir-reduction (PM) catalyst exhibited one significant peak higher than 800 °C, representing the reduction of Co(II) species that were strongly interacted with silica support. In contrast, almost no remarkable

Revised Fig. 2. (b) *in-situ* H₂-TPR profiles, (c) quasi *in-situ* Co 2p XPS of the four designed catalysts.

reduction peak was observed in the TPR curves of the Dir-reduction (PM200) and the Dir-reduction (PM900) catalysts, suggesting the absence of Co(II) species that were intensely interacted with SBA-15 support. Significantly, in the Dir-reduction (IMP900) catalyst, the characteristic reduction peak of embedded Co(II) sites on the support was observed, but the temperature and the intensity of the reduction peak at around 750 °C was downshift when compared to the Dir-reduction (PM) sample, revealing that Co species did not interact strongly with the support when the surface OH was insufficient. Furthermore, quasi *in-situ* XPS of the Dir-reduction (PM) catalyst from the revised Fig. 2c showed the BE of typical high-spin Co(II) species. On the contrary, the rest three catalysts indicated the peaks of metallic Co. In addition, samples were analyzed by the *ex-situ* UV-vis and recorded in the Fig. 2d (comment (6)). The absorbance at 543 nm, 578 nm, and 642 nm, as the characteristic band for Td-Co(II), were observed in the Dir-reduction (PM) and Dir-reduction (IMP900) catalysts. But the intensity of Td-Co(II) over the Dir-reduction (IMP900) was lower than that on the

Dir-reduction (PM). Over the Dir-reduction (PM200) and Dir-reduction (PM900) catalysts, the characteristic peak representing for cobalt oxide at approximately 410 nm and 720 nm were detected. It is worth noting that in the revised Fig. 2b, there was no reduction peak below 600 °C can be found, because these four catalysts were subjected to H₂-direct reduction treatment at 600 °C. However, since metallic cobalt was easily re-oxidized in air, *ex-situ* UV-vis spectra captured the diffraction peak of cobalt oxide (Fig. 2d in comment (6)).

We have revised our manuscript accordingly, and the reduction temperature of the contrast samples (PM200, PM900 and IMP900) has been presented in the revised manuscript (line 232-233, page 10; line 247-248, page 11). Moreover, other revisions can be found from the revised Fig. 2b, Fig. 2c (page 13) and the revised Manuscript (line 276-329, page 12-15).

Comment(8): The authors should explicit the function of the H₂ reduction process, because the authors say H₂ reduction process is to generate highly dispersed T_d-cobalt(II) sites in the abstract, while the authors also say that this process is to induce adsorb water in the line 106. Thus, the authors may need to explain it.

Reply: Thank you for the important suggestion. We feel sorry for the confusion caused by the unclear expression. This work aims to demonstrate the effect of adsorbed water for inducing highly dispersed tetrahedral cobalt(II). As for the direct H₂-reduction process, it is intended as a way to facilitate the acquisition of tetrahedral cobalt(II).

Previous studies have shown that support-stabilized T_d-cobalt(II) species resulted in the selective dehydrogenation of propane to propene, whereas bulk Co₃O₄, CoO, and metallic Co caused severe side reactions of cracking and coking. Thus, a variety of

synthetic strategies have been attempted to generate T_d-cobalt(II) sites. In a study by Yun's group (*J. Catal.* 2021, 404, 1007-1016), it was determined that H₂-treated Co/Al₂O₃ resulted in superior performance when compared to O₂ and Ar-treated catalysts. This is attributed to high ratio of T_d-cobalt(II) were stabilized on Al₂O₃. Therefore, we choose the direct H₂-reduction method to treat the catalysts. As a matter of fact, ligand-protected direct hydrogen reduction method has been widely developed in the synthesis of high performance catalyst (*Angew. Chem.* 2020, 132, 2-11), but the special role of this direct hydrogen reduction method was not clarified in detail.

In our work, the section of "Verifying the effect of adsorbed water on T_d-Co(II) formation" illustrated the role of adsorbed water for inducing highly dispersed tetrahedral cobalt(II) sites in detail. The direct H₂-reduction method was applied to the four contrast samples of Dir-reduction (PM), Dir-reduction (PM200), Dir-reduction (PM900) and Dir-reduction (IMP900). It is noted that when adsorbed water was absent, there was no way to obtain highly dispersed T_d-Co(II). Hence, the necessary condition for the availability of highly dispersed T_d-cobalt(II) sites was the presence of adsorbed water, rather than the direct H₂-reduction treatment.

We have revised our manuscript accordingly, and details can be found in line 516-521, page 23 of the revised manuscript.

Comment(9): In addition, it is better to enlarge the font size of some icons, like Fig. 1(a, b, c, d, e, f).

Reply: Thank you for pointing this out. As suggested by the reviewer, **we have enlarged the font size of icons in all the figures.**

Comment(10): Please specify the manufacturers and purity of all the chemicals and materials used in this work.

Reply: Thanks for your comments. Accordingly, **the manufacturers and purity of**

all the chemicals and materials used in this work are provided in the revised Supporting Information (line 27-30, page 2).

REVIEWERS' COMMENTS

Reviewer #3 (Remarks to the Author):

The detailed explanation and modifications have witnessed the authors' conscientious hardworking. This manuscript may be acceptable for publication after following minor revision.

In question 4, the meaning of "The notes of Fig. 1c, " $\text{Co}_3\text{O}_4 \rightarrow \text{CoO} \rightarrow \text{Co}$ " and " Co^{2+} ", should be unified" seems to be misunderstood. What I wanted to deliver is changing " Co^{2+} " to the form of " $\text{Co}^{2+} \rightarrow \text{Co}$ ", in the figure.

REVIEWER COMMENTS

Reviewer #3 (Remarks to the Author):

The detailed explanation and modifications have witnessed the authors' conscientious hardworking. This manuscript may be acceptable for publication after following minor revision. In question 4, the meaning of "The notes of Fig. 1c, " $\text{Co}_3\text{O}_4 \rightarrow \text{CoO} \rightarrow \text{Co}$ " and " Co_2^+ ", should be unified" seems to be misunderstood. What I wanted to deliver is changing " Co_2^+ " to the form of " $\text{Co}_2^+ \rightarrow \text{Co}$ ", in the figure.

Reviewer #3

The detailed explanation and modifications have witnessed the authors' conscientious hardworking. This manuscript may be acceptable for publication after following minor revision. In question 4, the meaning of “The notes of Fig. 1c, “ $\text{Co}_3\text{O}_4 \rightarrow \text{CoO} \rightarrow \text{Co}$ ” and “ Co^{2+} ”, should be unified” seems to be misunderstood. What I wanted to deliver is changing “ Co^{2+} ” to the form of “ $\text{Co}^{2+} \rightarrow \text{Co}$ ”, in the figure.

Reply: Thanks for your reminding. According to your valuable suggestions, we have made corresponding modifications to Fig. 1c.

Fig.1. Functional characterization of Co/SBA-15 catalysts. (c) H₂-TPR profiles